# G2T-LLM: Graph-to-Tree Text Encoding for Molecule Generation with Fine-Tuned Large Language Models

## Abstract

We introduce G2T-LLM, a novel approach for molecule generation that uses graph-to-tree text encoding to transform graph-based molecular structures into a hierarchical text format optimized for large language models (LLMs). This encoding converts complex molecular graphs into tree-structured formats, such as JSON and XML, which LLMs are particularly adept at processing due to their extensive pre-training on these types of data. By leveraging the flexibility of LLMs, our approach allows for intuitive interaction using natural language prompts, providing a more accessible interface for molecular design. Through supervised fine-tuning, G2T-LLM generates valid and coherent chemical structures, addressing common challenges like invalid outputs seen in traditional graph-based methods. While LLMs are computationally intensive, they offer superior generalization and adaptability, enabling the generation of diverse molecular structures with minimal task-specific customization. The proposed approach achieved comparable performances with state-of-the-art methods on various benchmark molecular generation datasets, demonstrating its potential as a flexible and innovative tool for AI-driven molecular design.

## 1 Introduction

Molecular generation is a critical task in fields such as drug discovery, material science, and chemistry (Schneider & Fechner, 2005; Simonovsky & Komodakis, 2018; Elton et al., 2019). The ability to design and create novel molecules with specific properties can accelerate the development of new therapies, advanced materials, and innovative chemicals. Traditional approaches to molecular generation, such as rule-based systems (Schneider & Fechner, 2005; Sastry et al., 2011) and graph-based (You et al., 2018; Madhawa et al., 2019; Shi et al., 2020) models, have provided foundational tools. However, these methods often face limitations in generating diverse, valid, and chemically coherent molecular structures, restricting their ability to explore the vast chemical space effectively (Vignac et al., 2022; Jo et al., 2022). Recent advancements in deep learning, especially the rise of large language models (LLMs), offer new opportunities for molecular generation (Brahmavar et al., 2024; Wang et al., 2024; Yao et al., 2024). Unlike traditional methods, LLMs are not constrained by domain-specific rules and can generalize from vast amounts of data. This flexibility allows them to generate creative and diverse content, potentially uncovering novel chemical compounds. Prior non-LLM approaches, such as graph-based generative models (You et al., 2018; Madhawa et al., 2019; Shi et al., 2020; Luo et al., 2021; Vignac et al., 2022; Jo et al., 2022), often struggle with limited generalization, rule-based rigidity, or difficulty scaling to more complex chemical structures. In contrast, LLMs can adapt to a wide range of prompts and provide greater flexibility, making them an attractive choice for AI-driven molecular generation.

Despite the promise of LLMs, applying them to molecular generation presents a unique challenge. Molecular structures are typically represented as graphs, with atoms as nodes and bonds as edges. LLMs, however, are trained to understand sequences of tokens (Vaswani, 2017), particularly in structured text formats such as XML and JSON (Brown, 2020), and are not inherently designed to process graph-based data. This mismatch creates a barrier when attempting to use LLMs for tasks that require understanding the relational and non-linear properties of molecular structures. LLMs (Luo et al., 2023; Le et al., 2024) may struggle to generate chemically valid or meaningful molecules without proper representation.

To overcome this challenge, we propose a novel Graph-to-Tree Text Encoding designed to transform molecular graphs into a format that LLMs can process effectively. Inspired by SMILES but not relying on it, our encoding converts graph-based molecular structures into hierarchical text representations, such as JSON and XML. These formats are naturally suited to LLMs, which excel at interpreting tree-like structures due to their training on similar data. By converting molecular graphs into tree-structured text, we align the data representation with the strengths of LLMs, enabling them to understand and generate molecules more effectively. With the graph-to-tree text encoding in place, we supervised fine-tuned LLMs to generate valid and coherent chemical structures. This fine-tuning process ensures that the generated molecules adhere to chemical rules and constraints, addressing common challenges such as the generation of invalid or chemically infeasible molecules. The fine-tuning allows LLMs to learn how to translate natural language prompts into meaningful molecular designs, opening new possibilities for human-guided molecule generation. Our approach has demonstrated comparable performances with state-of-the-art (SOTA) models on several benchmark molecular generation datasets. These results validate the effectiveness of our graph-to-tree encoding in making LLMs capable of generating chemically sound and diverse molecules. Additionally, the performance gains achieved underscore the potential of LLMs as a flexible and innovative tool for molecular generation, particularly when paired with a well-suited encoding.

This work makes the following contributions:

- We propose G2T-LLM, a novel approach that transforms graph-based molecular structures into text formats like JSON and XML, optimized for large language models.

- We introduce a token constraining technique to guide the LLM's generation process, ensuring that the output adheres to the expected tree-structured format, which is critical for maintaining molecular coherence.

- We develop a supervised fine-tuning method to enable LLMs to generate valid and coherent chemical structures, leveraging graph-to-tree text encoding.

- We achieve comparable performances with state-of-the-art models on benchmark molecular generation datasets, demonstrating the effectiveness and potential of our approach for AI-driven molecular design.

## 2 RELATED WORK

**Graph Generation.** The graph generation task aims to learn the distribution of graphs. The traditional approaches (Zang & Wang, 2020; Shi et al., 2020; Luo et al., 2021; You et al., 2018; Madhawa et al., 2019; Dai et al., 2018) such as auto-regression, Generative Adversarial Network (GAN), and Variational Autoencoder (VAE) have been explored for this purpose. However, they have faced challenges in modeling the permutation-invariant nature of graph distribution and learning the relationship between edges and nodes, often due to limitations in their model capacity. Recent advancements in diffusion methods (Niu et al., 2020; Jo et al., 2022; Vignac et al., 2022; Jo et al., 2023) have significantly improved graph generation. GDSS (Jo et al., 2022) generates both node features and adjacency matrices simultaneously, resulting in better alignment with graph datasets. DiGress (Vignac et al., 2022) addresses the challenge of generating graphs with categorical node and edge attributes, which is a difficult task due to the unordered nature and sparsity of graphs. GruM (Jo et al., 2023) directly learns graph topology, improving connectivity and structure recovery.

**Graph to Text for LLM.** The emergence of large language models (LLMs) has driven significant advancements in the natural sciences (Taylor et al., 2022; Liu et al., 2024). These models are trained on vast amounts of text data, the most abundant type of data, contributing to their success across many tasks. Multi-modal methods (Luo et al., 2023; Le et al., 2024) have been proposed to incorporate both graph and text information. They typically rely on graph neural networks or transformers to encode graphs. However, these methods often use text, such as SMILES, to represent molecular features. SMILES may not tokenize the molecular structure effectively, limiting the ability to represent the molecule structure accurately. As a result, the graph embeddings may be too weak for intricate molecular structures, limiting performance in molecular generation tasks.

Recently, there have been attempts (Fatemi et al., 2023) to represent graphs in natural language formats, encoding their structure using descriptive language. However, this naive approach introduces challenges, as such encodings are unlikely to appear in typical text, meaning that LLMs—trained

Figure 1: Illustration of the Graph-to-Tree Text Encoding process described in Section 3.2 and Algorithm 1. This figure shows how the molecular structure of cyclopropene is transformed into a hierarchical tree representation. Each atom and bond is mapped to nodes and edges in the tree, with unique identifiers assigned.

predominantly on conventional text data—may struggle to process them effectively. Using an encoding that aligns with the LLMs' training data is essential. We propose leveraging tree-structured formats like JSON and XML to encode molecules to address this issue. The JSON format is a widely used and structured data representation commonly found in LLM training. This allows us to capture the complexity of molecular graphs while ensuring compatibility with LLMs.

## 3 G2T-LLM

This section introduces G2T-LLM: Graph-to-Tree Text Encoding for Molecule Generation with Fine-Tuned Large Language Models.

### 3.1 CHALLENGES AND MOTIVATIONS

Molecular graphs pose a challenge for LLMs due to their inherently complex, non-linear structures, where atoms (nodes) and bonds (edges) form intricate connectivity patterns, including rings, branches, and cycles. Traditional LLMs excel at processing sequential data, such as natural language, where information flows in a linear manner. However, molecular graphs do not naturally conform to this format, as their connections often lack a clear, ordered sequence. This mismatch complicates the application of LLMs to molecule-related tasks.

Despite these challenges, LLMs have shown a capacity to handle structured, hierarchical data formats, such as JSON and XML. These formats share some of the complexity of graphs but are still expressed as trees, with clear parent-child relationships between elements. LLMs trained on such data can handle hierarchical structures by processing them as sequences while maintaining the relationships and nested dependencies inherent to these structures. This training has made LLMs particularly adept at handling data that can be decomposed into nested layers, making them better suited for tree-like representations than arbitrary graphs.

To leverage this strength, we propose encoding molecular graphs into a tree structure. This approach is inspired by SMILEs, which are essentially tree representations of molecular graphs, proving that molecular graphs can be effectively serialized as trees while preserving their chemical properties. This encoding acts as a bridge between the graph-based molecular structures and the LLM's ability to process and generate hierarchical data. The LLM can be trained on these tree-encoded molecules, and it can also output molecules in the same structured format, facilitating the generation of coherent molecular representations. By aligning graph data with a format that LLMs are well-equipped to handle, this method holds the potential for improving the coherence and plausibility of generated molecular structures.

**Algorithm 1** Convert Molecular Graph to Tree-Structured Text Representation

```
 1: function GRAPH2TREE(graph)
 2:     Input: graph (dictionary of atom identifiers to connected atom identifiers)
 3:     Output: text_representation (tree structure in text format)
 4:     tree ← {}                                                      ▷ Initialize tree
 5:     visited ← {}                                          ▷ Set to track visited atoms
 6:     unique_id_counter ← 0                               ▷ Counter for unique atom IDs
 7:     id_mapping ← {}                                   ▷ Mapping of atoms to unique IDs
 8:     function CONVERTATOM(atom)
 9:         visited.add(atom)
10:         atom_id ← unique_id_counter
11:         id_mapping[atom] ← atom_id
12:         unique_id_counter ← unique_id_counter + 1
13:         bonds ← []
14:         for neighbor, bond_type in graph[atom] do
15:             if neighbor ∉ visited then
16:                 child ← CONVERTATOM(neighbor)
17:             else
18:                 neighbor_id ← id_mapping[neighbor]
19:                 child ← {"atom_name": atom.atom_name, "atom_id": neighbor_id, "bonds": []}
20:                                         ▷ Set bonds to empty to avoid circular references
21:             bonds.append({"atom": child. "bond_type": bond_type})
22:         return {"atom_name": atom.atom_name, "atom_id": atom_id, "bonds": bonds}
23:     root_atom ← any(graph.keys())                       ▷ Start from any atom as the root
24:     tree ← CONVERTATOM(root_atom)
25:     text_representation ← JSON.stringify(tree)              ▷ Convert tree to JSON text format
26:     return text_representation
```

## 3.2 GRAPH-TO-TREE TEXT ENCODING

To make molecular graphs accessible to LLMs, we introduce a tree-based encoding inspired by the SMILES format. SMILES encodes molecules by performing a depth-first traversal over the molecular graph and representing it as a linear string. In our approach, we extend this traversal to build a hierarchical tree structure, where atoms are represented as nodes and their bonds as edges connecting them. The hierarchical nature of the tree is well-suited for the LLM's training with tree-like structures.

However, molecular graphs often contain rings and cycles—features that trees cannot naturally represent. To address this, we assign each atom in the molecule a unique identifier (ID). When the traversal encounters a ring closure or cycle, the tree refers back to the atom's unique ID rather than creating a new node, thereby preserving both the hierarchical structure and chemical validity. This encoding technique ensures that we accurately capture the full molecular graph in a way the LLM can process, while maintaining the integrity of complex molecular features such as rings and branches. Algorithm 1 and Algorithm 2 describe the processes for converting a molecular graph to a tree-structured text representation and for reconstructing the graph from this format, respectively. Figure 3 illustrates the graph-to-tree text encoding.

## 3.3 TOKEN CONSTRAINING FOR VALID TREE-STRUCTURE GENERATION

Despite the advancements in LLMs, there remains a significant challenge in ensuring that the outputs adhere to valid tree-structured formats. LLMs, while capable of generating coherent text, may produce sequences that do not respect the hierarchical relationships required for molecular representation. This can lead to outputs that are structurally invalid, failing to accurately represent the complex relationships inherent in molecular graphs.

To mitigate this issue, we implement a set of constraints that guide the token generation process of the LLM. These constraints filter the tokens allowed at each step, ensuring that generated outputs remain within the bounds of valid tree structures. Specifically, we impose rules that dictate accept-

---

**Algorithm 2** Convert Tree-Structured Text to Molecular Graph

---

1: **function** TREE2GRAPH(tree_json)
2:     **Input:** tree_json (tree structure in JSON format)
3:     **Output:** graph (dictionary representing the molecular graph)
4:     tree ← JSON.parse(tree_json)                      ▷ Convert JSON text to tree structure
5:     graph ← {}                                        ▷ Initialize graph structure
6:     **function** CONVERTNODETOGRAPH(node, parent, bond_type)
7:         atom_id ← node["atom_id"]
8:         **if** atom_id ∈ id_mapping **then**
9:             atom ← id_mapping[atom_id]
10:         **else**
11:             atom_name ← node["atom_name"]
12:             atom ← new Node(atom_name)
13:             id_mapping[atom_id] ← atom
14:             graph[atom] ← []                          ▷ Initialize adjacency list
15:         **if** parent_id ≠ null **then**
16:             graph[atom].append((parent, bond_type))
17:             graph[parent].append((atom, bond_type))
18:         **for** child **in** node["bond"] **do**
19:             CONVERTNODETOGRAPH(child, atom)
20:     root_node ← tree                                  ▷ Start with the root node of the tree
21:     CONVERTNODETOGRAPH(root_node, null)
22:     **return** graph

---

able parent-child relationships, enforce valid connections between atoms, and restrict the formation of non-hierarchical sequences. Additionally, we constrain the types of atoms and bonds that can be generated, ensuring that only valid atom types (e.g., carbon, oxygen) and bond types (e.g., single, double) are used in the output. This approach leverages domain knowledge of molecular structures to create a robust framework for guiding the LLM's outputs.

The application of token constraining significantly enhances the reliability of the generated tree-structured outputs. By enforcing these constraints, we improve the chances that the LLM produces valid representations of molecular structures that can be effectively used in further analyses or applications. This technique not only aids in ensuring the accuracy of the generated data but also reinforces the overall effectiveness of our graph-to-tree text encoding approach, making it a vital component in achieving coherent and chemically valid molecular generation.

### 3.4    SUPERVISED FINE-TUNING LLMs FOR MOLECULAR GENERATION

A key challenge in leveraging large language models for molecular generation is that, without specialized training, they may struggle to produce valid molecular structures, particularly when dealing with complex features such as rings, cycles, and the inherent chemical constraints that govern molecular formation. Supervised fine-tuning addresses this issue by teaching the LLM domain-specific rules and patterns, enabling it to generate valid molecular structures that adhere to chemical principles.

We structure the fine-tuning process as a molecular completion task. The LLM is trained by prompting it with a partial molecular structure, encoded using the graph-to-tree text encoding and tasking it with predicting the remaining atoms and bonds necessary to complete the molecule. For each training example, we provide the LLM with an incomplete molecular graph, and the model is then expected to generate the missing parts based on the information provided. The model's output is evaluated against the full molecular structure's text encoding, and the loss is computed based on the accuracy of its predictions. By iterating through this process, the LLM learns to predict the completion of molecular graphs in a way that respects chemical validity, helping the model better handle challenging structural features. Note that token constraining is deliberately omitted during fine-tuning, allowing the LLM to explore and learn more freely before constraints are imposed during inference. Figure 3.4 illustrates the supervised fine-tuning process of G2T-LLM.

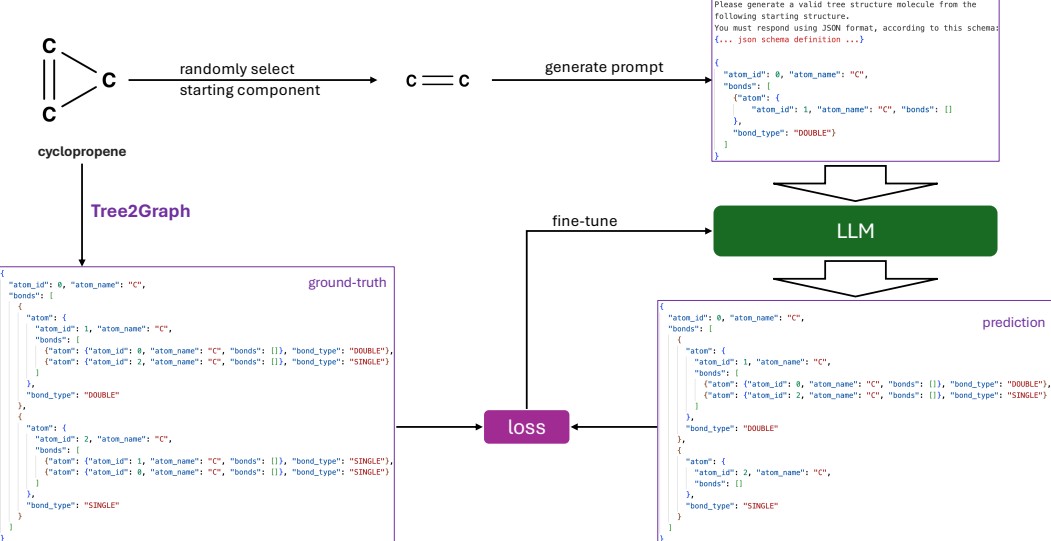

Figure 2: An illustration of the supervised fine-tuning process of G2T-LLM. The process begins by randomly selecting a starting component, exemplified by cyclopropene, which is encoded into a partial tree structure and passed as a prompt to the LLM. The LLM generates the remaining molecular structure, which is compared against the ground truth. A loss is computed and is used to fine-tune the model, iteratively improving its performance in generating valid molecular graphs.

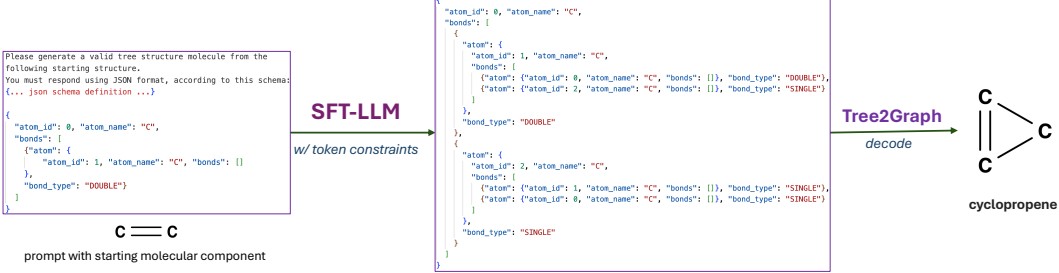

Figure 3: An illustration of the inference process of G2T-LLM. The process starts by prompting the model with a random molecular component. The model, a fine-tuned LLM (SFT-LLM), generates new molecular structures while applying token constraints to ensure valid outputs. The output is a tree-structured text representing the molecule. It is then decoded back into a molecular graph corresponding to cyclopropene.

The fine-tuning process is integral to the success of our approach. By casting molecular generation as a completion task and using the proposed graph-to-tree encoding as a bridge between molecular structures and the LLM's capabilities, we enhance the model's ability to generate coherent and chemically valid outputs. This fine-tuning approach refines the LLM's understanding of molecular patterns and constraints, enabling it to produce outputs that are more reliable and scientifically grounded within the realm of molecular design.

## 3.5 INFERENCE PROCESS OF G2T-LLM

The molecular generation process begins with selecting a random molecular component, which could be an atom, a bond, or even a larger motif. This component serves as the initial prompt for the fine-tuned LLM. The component is encoded into the graph-to-tree text format, creating a tree-structured representation that the LLM can process.

Once the LLM receives this initial prompt, it is tasked with generating the subsequent components of the molecular structure. At each step, the LLM's output is constrained by the Token Constraining mechanism, ensuring that only chemical and schema-valid tokens—such as specific atom types and bond types—are generated. These constraints help guide the LLM in maintaining the coherence of the structure, preventing invalid or nonsensical outputs, and ensuring that the generated molecule adheres to the expected chemical rules. As the LLM iteratively predicts new components, these outputs are progressively combined into an expanding tree-structured text. This generated text represents the molecular graph, with nodes corresponding to atoms and edges corresponding to bonds. Once the generation process is complete, the final tree-structured text is decoded back into a full molecular graph. This graph is then translated into a standard molecular format, fully reconstructing the molecule from the text generated by the LLM. Figure 3.4 illustrates the inference process of G2T-LLM.

## 4 EXPERIMENTS

In this section, we conduct comprehensive experiments on two real-world datasets to evaluate the effectiveness of our proposed methods.

### 4.1 EXPERIMENTAL SETUP

**Datasets and Metrics.** We evaluate the quality of molecule generation using two real-world datasets: *QM9* (Ramakrishnan et al., 2014) and *ZINC250k* (Irwin et al., 2012). Following the evaluation setting used in (Jo et al., 2023), we measure model performance across four metrics. *Validity* is the proportion of generated molecules that are valid without any valency corrections. *Novelty* is the proportion of valid molecules that are not present in the training dataset. *Frechet ChemNet Distance (FCD)* (Preuer et al., 2018) measures the similarity between two molecule sets by comparing the activations of the penultimate layer of the ChemNet model. *Scaffold similarity (Scaf.)* evaluates the model's ability to generate similar substructures.

**Baselines.** We compare our model with following molecular graph generation methods. *MoFlow* (Zang & Wang, 2020) is a one-shot flow-based model that generates entire molecular graphs in a single step. *GraphAF* (Shi et al., 2020) and *GraphDF*(Luo et al., 2021) are autoregressive flow-based models, generating molecules sequentially. Additionally, we evaluate against the diffusion models. *EDP-GNN* (Niu et al., 2020) is a score-based model designed for generating adjacency matrices. *GDSS* (Jo et al., 2022) uses a continuous diffusion process for molecule generation, *DiGress* (Vignac et al., 2022) employs a discrete diffusion approach, and *Grum* (Jo et al., 2023) designed a mixture of endpoint-conditioned diffusion processes.

Although several studies have explored using LLMs for molecular generation, direct comparisons with our approach are not feasible. For instance, LMLF (Brahmavar et al., 2024), Grammar Prompting (Wang et al., 2024), and LLM4GraphGen (Yao et al., 2024) all employ rule-based prompt-engineering techniques that fundamentally differ from our SFT LLM approach. These models rely on predefined rules and heuristics to guide the generation process, which restricts their ability to learn from the underlying data distributions. In contrast, our method leverages a more flexible and adaptive encoding, allowing the LLM to capture the complexities of molecular structures more effectively.

Moreover, the baseline models utilize significantly larger architectures, such as GPT-4, whereas our experiments are conducted with LLaMA3.1-8B. This disparity in model size and complexity further complicates direct comparisons, as the performance capabilities and learned representations of these models can vary widely. Therefore, assessing our results against those achieved by larger, rule-based models may not provide a meaningful evaluation of performance, given the substantial differences in methodologies and model architectures.

**Implementation details.** For our G2T-LLM, we conduct experiments using the LLaMA3.1-8B model (Dubey et al., 2024) as our base LLM, selected for its strong performance in text generation tasks. The model parameters are fine-tuned with torchtune (Ansel et al., 2024), and we leverage QLoRA (Dettmers et al., 2024) to accelerate training while reducing memory consumption. The fine-tuning dataset consists of 5,000 molecules, and the model is trained with a batch size of 8, using the AdamW optimizer (Loshchilov, 2017) with a weight decay of 0.01 and a learning rate of

Table 1: Generation results on the QM9 and ZINC250k datasets. We report the mean of 3 different runs. The best results are highlighted in bold. The second-best results are highlighted in underline. We provide the results of uniqueness, and NSPDK in Appendix A.

| Datasets | QM9 | | | | ZINC250K | | | |
|---|---|---|---|---|---|---|---|---|
| Methods | Valid↑ | Novelty↑ | FCD↓ | Scaf↑ | Valid↑ | Novelty↑ | FCD↓ | Scaf↑ |
| MoFlow | 91.36 | 94.72 | 4.467 | 0.1447 | 63.11 | **100.00** | 20.931 | 0.0133 |
| GraphAF | 74.43 | 86.59 | 5.625 | 0.3046 | 68.47 | 99.99 | 16.023 | 0.0672 |
| GraphDF | 93.88 | **98.54** | 10.928 | 0.0978 | 90.61 | **100.00** | 33.546 | 0.0000 |
| EDP-GNN | 47.52 | 86.58 | 2.680 | 0.3270 | 82.97 | **100.00** | 16.737 | 0.0000 |
| GDSS | 95.72 | 86.27 | 2.900 | 0.6983 | 97.01 | **100.00** | 14.656 | 0.0467 |
| DiGress | 98.19 | 25.58 | **0.095** | 0.9353 | 94.99 | 99.99 | 3.482 | 0.4163 |
| Grum | **99.69** | 24.15 | 0.108 | **0.9449** | **98.65** | 99.98 | **2.257** | 0.5299 |
| Ours | 99.47 | 88.29 | 0.815 | 0.9112 | 98.03 | **100.00** | 2.445 | **0.6062** |

Figure 4: Visualization of the generated molecules with Tanimoto similarity scores based on Morgan fingerprints. The best results are highlighted in bold.

3e-4. The learning rate is adjusted by a cosine schedule with 100 warmup steps, and cross-entropy loss is employed for the loss computation. All model computations are performed with the bf16 data type. Fine-tuning is carried out on an NVIDIA A100 SXM4 80GB, and inference is done on NVIDIA GeForce RTX 3090 and 4090 GPUs. The implementation is done in PyTorch (Paszke et al., 2019).

## 4.2 EXPERIMENTAL RESULTS

Table 1 presents performance comparisons on both the QM9 and ZINC250k datasets against baseline models. Our approach consistently achieves top-two validity scores across both datasets, demonstrating its effectiveness in enabling the LLM to capture the underlying chemical rules essential

for accurate molecule generation. For novelty, our method attains a perfect score of 100% on the ZINC250k dataset and 88% on QM9, highlighting its ability to consistently generate novel molecular structures. In terms of FCD and Scaf metrics—critical indicators of a model's ability to explore and replicate chemical space—our method delivers competitive performance compared to other baselines. While DiGress and Grum show strong FCD and Scaf scores on the QM9 dataset, their novelty scores fall significantly short (below 40%), suggesting potential overfitting to the training data rather than true generalization of molecular distributions. In contrast, our method not only maintains high novelty rates but also achieves strong performance on FCD and Scaf metrics. On the ZINC250k dataset, our approach attains the highest Scaf score and the second-best FCD score, further demonstrating its superior ability to generalize and innovate within chemical spaces. This robust performance underscores our model's advanced understanding and application of molecular distributions, making it a powerful tool for innovative molecular design in computational chemistry.

## 4.3 VISUALIZATION RESULTS OF GENERATED MOLECULES

In Fig. 4, we follow the experimental setup outlined in (Jo et al., 2022), using Tanimoto similarity based on Morgan fingerprints to evaluate the generated molecular graphs. For consistency and comparability, we select the same molecules as (Jo et al., 2022). Additionally, we perform experiments on molecular graphs generated by Grum (Jo et al., 2023). Across most cases, our method demonstrates superior performance compared to previous state-of-the-art diffusion-based approaches, showcasing its effectiveness and robustness in molecular graph generation.

## 4.4 ABLATION STUDY: IMPACT OF TREE-STRUCTURED TEXT ENCODING

To evaluate how our proposed graph-to-tree text encoding improves the LLM's ability to learn graph structures compared to the previous graph-to-text methods such as Talk Like a Graph (Fatemi et al., 2023), we conducted experiments on the challenging Zinc250K dataset (Irwin et al.,

Table 2: Study of the impact of tree-structured text encoding on the ZINC250K dataset.

| Methods | Valid↑ | FCD↓ | Scaf↑ | Novelty↑ |
|---|---|---|---|---|
| Talk like a graph | 59.20 | 19.8114 | 0.1317 | 100 |
| Ours | 98.60 | 5.6906 | 0.1522 | 100 |

2012), which contains larger molecules. Talk Like a Graph encodes graph structures by converting them into natural language, where each node's connections and attributes are described in sentence form. For the fine-tuning process, we randomly selected 5,000 molecules from the training set and generated 1,000 molecules for performance comparison. As shown in Table 2, our method significantly outperforms the previous approach across all metrics, demonstrating that encoding molecular structures in JSON format enables LLMs to more effectively learn and replicate complex molecular structures.

## 4.5 ABLATION STUDY: IMPACT OF SUPERVISED FINE-TUNING LLM

In this study, we aim to evaluate the impact of supervised fine-tuning on LLM performance. Specifically, we generate 1,000 molecules using the same prompt to compare the performance of the LLM before and after fine-tuning. This direct comparison allows us to assess how fine-tuning enhances the model's ability to accurately generate molecular structures. We conduct this experiment using the ZINC250k

Table 3: Comparison of LLM performance with and without SFT on the ZINC250k dataset.

| Methods | Valid↑ | Unique↑ | Novelty↑ |
|---|---|---|---|
| w/o SFT | 70.80 | 61.12 | 100.00 |
| w/ SFT | 98.60 | 98.98 | 100.00 |

dataset, and the results are presented in Table 3. The results reveal that without fine-tuning, the LLM produces molecules with only 70.8% validity and 61.12% uniqueness, indicating that the model, in its initial state, struggles to fully comprehend and accurately replicate the text representation of molecular structures. However, after fine-tuning, there is a significant improvement, with validity and uniqueness increasing to 99.6% and 99.79%, respectively. These results highlight the effectiveness of fine-tuning in substantially improving the model's performance, demonstrating its critical role in enabling the LLM to better understand and generate precise molecular structures.

### 4.6 ABLATION STUDY: IMPACT OF SIZE OF THE FINE-TUNING DATASET

In this section, we investigate the impact of dataset size on the performance of a LLM during fine-tuning. Our experiments use the QM9 dataset with three distinct dataset sizes for fine-tuning: 1,000, 5,000, and 10,000 molecules. Each model is trained over 10 epochs. This setup enables a systematic evaluation of how variations

Table 4: Comparison of LLM performance with different size of fine-tuning datasets

| Methods | Valid↑ | Novelty↑ | FCD↓ | Scaf↑ |
|---|---|---|---|---|
| 1k (10 epoch) | 98.50 | 90.38 | 1.226 | 0.6933 |
| 5k (10 epoch) | 98.70 | 86.53 | 1.219 | 0.7779 |
| 10k (10 epoch) | 98.50 | 73.89 | 1.146 | 0.7980 |

in fine-tuning data size affect the model's learning efficacy and its ability to generalize. Table 4 presents the results of these experiments. The results indicate an improvement in the FCD and Scaf scores as the dataset size increases. This improvement likely stems from the LLM's exposure to a larger array of data points, which enhances its understanding of the chemical distribution within the dataset. Conversely, we observe a decrease in novelty scores with larger datasets. This reduction may be attributed to the relatively small and structurally simple nature of the QM9 dataset, which comprises only four types of atoms and molecules not exceeding nine atoms. As the model encounters more data, it increasingly reproduces similar outputs, reflecting the limited diversity in the dataset.

### 4.7 ABLATION STUDY: IMPACT OF TOKEN CONSTRAINING

In this section, we examine the impact of token constraining on molecular generation, as introduced in Section 3.3. Token constraining is implemented to guide the LLM toward generating valid molecular structures by restricting its output to adhere to chemical rules. To evaluate the effectiveness of this approach, we perform an experimental comparison using the ZINC250k

Table 5: Comparison results of using token constraining (TC) on molecular generation on the ZINC250k dataset.

| | w/o TC | w/ TC |
|---|---|---|
| Validity (%)↑ | 41.60 | 98.60 |

dataset. Specifically, we generate 1,000 molecules to compare the validity of the output with and without token constraining. The results, presented in Table 5, clearly demonstrate the efficacy of token constraining in improving the validity of generated molecules. Without token constraining, the validity of the generated molecules is only 41.6%. However, when token constraining is applied, validity dramatically increases to 98.6%. This significant improvement underscores the critical role of token constraining in guiding the LLM to produce valid molecular structures, ensuring closer adherence to the fundamental rules of chemical structure and leading to a higher rate of valid outputs.

## 5 CONCLUSION

In this work, we introduced G2T-LLM, a novel approach for molecular generation that leverages LLMs to generate valid molecular structures through a novel graph-to-tree text encoding. By converting molecular graphs into hierarchical representations inspired by SMILES but adapted for LLMs, we bridge the gap between non-linear molecular structures and sequential data processing. This encoding allows the LLM to understand the molecular structure better and produce coherent chemical outputs. Our method addresses the challenges of generating valid molecular structures by introducing token constraints during the generation process, ensuring that the outputs respect some chemical and structural rules. Through supervised fine-tuning, we further align the LLM with molecular generation tasks, improving its ability to produce chemically valid molecules based on the learned data patterns from benchmark datasets like Zinc250K and QM9. Our results demonstrate the effectiveness of G2T-LLM, achieving state-of-the-art performance on benchmark datasets. This work highlights the potential of utilizing LLMs in molecular design, opening up new avenues for AI-driven discoveries in chemistry. The combination of hierarchical encoding, token constraining, and fine-tuning proves to be a powerful strategy for tackling the complexities of molecular generation. Future work will focus on refining these techniques to enhance efficiency and explore further applications in drug discovery and material science.

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

## A    ADDITIONAL EXPERIMENTS RESULTS

Here are additional experiment results on QM9 and ZINC250k datasets. The **Neighborhood Subgraph Pairwise Distance Kernel (NSPDK) Maximum Mean Discrepancy (MMD)** (Costa & Grave, 2010) evaluates the difference between generated and test molecules, accounting for both node and edge features. **Uniqueness** refers to the percentage of valid molecules that are distinct from each other. **Validity**, **FCD**, **Novelty**, and **Scaf** have been introduced before.

Table 6: Generation results on the QM9 dataset. We report the mean of 3 different runs. The best results are highlighted in bold. The second-best results are highlighted in underline.

| Methods | Valid (%)↑ | FCD ↓ | NSPDK ↓ | Scaf ↑ | Unique (%)↑ | Novelty (%)↑ |
|---|---|---|---|---|---|---|
| MoFlow | 91.36 | 4.467 | 0.017 | 0.1447 | 98.65 | 94.72 |
| GraphAF | 74.43 | 5.625 | 0.021 | 0.3046 | 88.64 | 86.59 |
| GraphDF | 93.88 | 10.928 | 0.064 | 0.0978 | 98.58 | **98.54** |
| EDP-GNN | 47.52 | 2.680 | 0.005 | 0.3270 | **99.25** | 86.58 |
| GDSS | 95.72 | 2.900 | 0.003 | 0.6983 | 98.46 | 86.27 |
| DiGress | 98.19 | **0.095** | 0.0003 | 0.9353 | 96.67 | 25.58 |
| Grum | **99.69** | 0.108 | **0.0002** | **0.9449** | 96.90 | 24.15 |
| Ours | 99.47 | 0.815 | 0.002 | 0.9112 | 89.57 | 88.29 |

Table 7: Generation results on the ZINC250k dataset. We report the mean of 3 different runs. The best results are highlighted in bold. The second-best results are highlighted in underline.

| Methods | Valid (%)↑ | FCD ↓ | NSPDK ↓ | Scaf ↑ | Unique (%)↑ | Novelty (%)↑ |
|---|---|---|---|---|---|---|
| MoFlow | 63.11 | 20.931 | 0.046 | 0.0133 | **99.99** | **100.00** |
| GraphAF | 68.47 | 16.023 | 0.044 | 0.0672 | 98.64 | 99.99 |
| GraphDF | 90.61 | 33.546 | 0.177 | 0.0000 | 99.63 | **100.00** |
| EDP-GNN | 82.97 | 16.737 | 0.049 | 0.0000 | 99.79 | **100.00** |
| GDSS | 97.01 | 14.656 | 0.019 | 0.0467 | 99.64 | **100.00** |
| DiGress | 94.99 | 3.482 | 0.0021 | 0.4163 | 99.97 | 99.99 |
| Grum | **98.65** | **2.257** | **0.0015** | 0.5299 | 99.97 | 99.98 |
| Ours | 98.03 | 2.445 | 0.0049 | **0.6062** | 94.69 | **100.00** |

