# OpenReview forum: "G2T-LLM: Graph-to-Tree Text Encoding for Molecule Generation with Fine-Tuned Large Language Models"
_ICLR.cc/2025/Conference — ICLR 2025 Conference Withdrawn Submission_

### Official Review · Reviewer_rrFx · 2024-10-28

**Soundness:** 2
**Presentation:** 3
**Contribution:** 2
**Rating:** 3
**Confidence:** 5

**Summary:**

This paper introduces G2T-LLM, a method that integrates hierarchical encoding, token constraining, and supervised fine-tuning for the generation of small molecules. Hierarchical encoding involves converting graph-based molecular structures into hierarchical text formats, such as JSON and XML. Additionally, token constraining is employed to prevent the generation of invalid structures, while supervised fine-tuning focuses on generating hierarchical text based on the molecular substructure. This approach effectively produces valid and coherent molecules, demonstrating competitive performance against baseline methods.

**Strengths:**

- The authors propose a novel approach to converting molecular data using XML and JSON, an area that has not been extensively explored before.
- The paper presents various analyses and ablation studies that support the results.

**Weaknesses:**

- While molecular data conversion is indeed less explored, there have been a few works that have utilized XML or JSON for graph data applications in large language models (LLMs) [1]. Additionally, representing graphs in a tree format (e.g., nodes connected to each other) is fundamentally similar to the proposed hierarchical encoding [2]. Therefore, the novelty of this approach is somewhat limited.
- There is a lack of discussion regarding related works. The proposed method appears to be closely related to HGGT [3], which also presents a sequential tree-based representation. Incorporating this comparison in the experiments would strengthen the paper significantly.
- The paper does not adequately explain the advantages of the proposed representation compared to SMILES. The statement “SMILES may not tokenize the molecular structure effectively” (line 103) needs clarification. Essentially, both SMILES and the proposed hierarchical representation serve similar purposes, although SMILES may require additional steps to traverse nodes when encountering obstacles. Furthermore, SMILES can accurately represent detailed structures, including rings, aromatic rings, and branches, making it more informative than the proposed method. Including SMILES generation results in the experiments (e.g., by prompting for SMILES or using supervised fine-tuning for SMILES generation) could partially address this issue, but further explanation is required.
- The motivation for using JSON format or tree representation is not well-supported by relevant references. The paper claims that these formats are widely used in LLM training and that LLMs have demonstrated the capacity to handle hierarchical data formats, yet no supporting references are provided.
- The algorithm lacks sufficient detail. For example, the token constraining algorithm does not specify which invalid node and bond types are filtered out, how to enforce a valid JSON format, or how this algorithm integrates with the decoding scheme of the LLM. Since the authors used Llama, enforcing constraints at each step could be challenging. Additionally, the process for randomly selecting the starting component is not clearly explained.
- The results do not appear to show a significant improvement compared to the baselines.

[1]  Kerui Zhu, et al.,Investigating Instruction Tuning Large Language Models on Graphs, COLM 2024.

[2] Ruosong Ye, et al., Langage is all a graph needs. EACL 2024.

[3] Yunhui Jang, et al., Graph generation with K2-trees. ICLR 2024.

**Questions:**

- There seems to be duplicated information in the representation; for instance, a double bond is represented twice for the edges (0,1) and (1,0). Why is there no technique, such as pruning, applied? Since the molecular graph is undirected, this could simplify the representation.
- Why not train the tree-to-graph model directly instead of randomly selecting the starting component? What are the advantages of this approach? The random selection of starting points seems questionable, as any molecule containing the substructure could be considered a valid output, making it difficult to define a unique loss function.

---

### Official Review · Reviewer_Atf4 · 2024-10-29

**Soundness:** 1
**Presentation:** 1
**Contribution:** 1
**Rating:** 3
**Confidence:** 4

**Summary:**

Through this paper, the authors propose G2T-LLM, a method for molecule generation that uses graph-to-tree text encoding to transform graph-based molecular structures into a hierarchical text format optimized for LLMs. The proposed G2T-LLM achieved comparable performances with SOTA methods on unconditional molecule generation tasks.

**Strengths:**

- Ablation studies on the impact of the proposed text encoding and supervised LLM fine-tuning were conducted.

**Weaknesses:**

I will combine the *Weaknesses* section and the *Questions* section. My concerns are as follows:
- The Introduction and Related Work sections are not well organized. In Introduction, only rule-based and graph-based methods were mentioned as previous approaches. In Related Work, only graph generation and graph-to-text methods were introduced. However, sequence-based methods that represents molecules as SMILES [1] or SELFIES [2] are also a very important and big branch in molecular generation. The authors need to list the previous methods well and state the limitations of each method in detail.
- Overall, I think the figures and tables are not carefully drawn and do not help the reader understand. Figure 1, 2, and 3 are drawn with poor abstraction. The raw codes are just pasted, and the figures not readable at a glance. The authors could have used shapes and diagrams to better explain what is happening in each process. Algorithms 1 and 2 are also written with poor abstraction, and the variables and functions used are poorly explained. They cannot be called well-written pseudocode.
- Another main weakness of this paper is that the experimental setup is not sound and extensive enough. Previous papers like GDSS [3], DiGress [4], and GruM [5] are generic graph generation methods and therefore include generic graph generation experiments. On the other hand, recent papers focusing on molecule generation include goal-directed molecule generation benchmarks as well as unconditional molecule generation, with the latter being the main experiment [6, 7]. This is because unconditional molecule generation is less practical and less important for real-world drug discovery problems. To claim that this paper proposes a methodology that is specific to molecule generation, I strongly suggest to include the goal-directed molecule generation experiment as the main experiment. Moreover, sequence-based molecular generation models are missing as baselines.
- The authors described that QM9 is a real-world dataset on Line 344, which is incorrect: most of the molecules in QM9 are very small (those are organic molecules with up to 9 heavy atoms) and do not exist in the real world.
- What are the specific differences between G2T-LLM and SFT-LLM? There is no mention of SFT-LLM in the abstract or introduction, and the only mention of it is in the caption of Figure 3. It seems that the terms are being used interchangeably without a strict definition.
- The authors did not provide the codebase to reproduce the results.

---

**References:**

[1] Weininger, SMILES: a chemical language and information system, Journal of chemical information and computer sciences 28.1 (1988): 31-36.

[2] Krenn et al., Self-referencing embedded strings (SELFIES): a 100% robust molecular string representation, Machine Learning: Science and Technology 1.4 (2020): 045024.

[3] Jo et al., Score-based generative modeling of graphs via the system of stochastic differential equations, ICML, 2022.

[4] Vignac et al., DiGress: discrete denoising diffusion for graph generation, ICLR, 2023.

[5] Jo et al., Graph generation with diffusion mixture, ICML, 2024.

[6] Wang et al., Retrieval-based controllable molecule generation, ICLR, 2023.

[7] Lee et al., Drug discovery with dynamic goal-aware fragments, ICML, 2024.

---

**Typos:**
- Line 357 & Table 1 & Figure 4, Grum -> GruM
- Line 362, SFT LLM -> SFT-LLM
- Line 461 & Line 534, Zinc250k -> ZINC250k

**Questions:**

Please see the *Weaknesses* section for my main concerns.

---

### Official Review · Reviewer_F53n · 2024-10-31

**Soundness:** 2
**Presentation:** 3
**Contribution:** 2
**Rating:** 3
**Confidence:** 4

**Summary:**

This paper proposes G2T-LLM, which transforms the molecules as json format for LLM learning. The authors additionally propose a token constraining technique to generate the valid molecules. G2T-LLM achieves comparable performances with state-of-the-art models.

**Strengths:**

1. Transforming graphs as json format is interesting and the authors introduce a token constraining technique to guide the generation process.

**Weaknesses:**

1. The overall performance is actually not strong compared with the baselines.

2. The baselines or ablation study compared other textual methods to represent molecules, such as SMILES and SMARTS.

3. The json format compared with SIMLES is less straightforward and more time-consuming to generate. Besides, the json format ignore some inductive bias in molecules, such as valence.

**Questions:**

See weaknesses.

---

### Official Review · Reviewer_hiSx · 2024-11-01

**Soundness:** 2
**Presentation:** 2
**Contribution:** 2
**Rating:** 3
**Confidence:** 4

**Summary:**

The paper proposes an interesting method for molecule generation, which first represents the molecular graph with structured data in JSON format, and then uses a Large Language Model (LLM) to learn the patterns of structured molecules for molecule generation. To generate valid JSON-formatted molecules, the authors limit the token space generated by the LLM. The authors conducted experiments on the QM9 and ZINC250k datasets, demonstrating that their proposed model can generate valid and novel molecules.

**Strengths:**

The paper proposes an interesting method for molecule generation and validates its effectiveness. The paper is clearly written, and the experiments are comprehensive.

**Weaknesses:**

1. The conversion of molecular graphs into JSON is interesting, but the motivation and necessity are somewhat far-fetched. Molecules can not only be represented by graphs but also by serialized representations, such as IUPAC, SMILES, InChl. These are widely recognized molecular serialization representation methods in the biological field. These representation methods can also be recognized by LLM, and they are superior to JSON in terms of readability, convenience, and complexity. The authors propose a new way of representing molecules and should elaborate on where this method is superior to traditional methods.

2. The authors state that graph-based methods often face limitations in generating diverse, valid, and chemically coherent molecular structures, restricting their ability to explore the vast chemical space effectively. However, from the experimental results, the method proposed by the authors does not have an advantage over graph-based methods. For example, on QM9, GraphDF's validity is 93.88, novelty is 98.54; the proposed method's validity is 99.48, novelty is 88.29; although validity has increased by about 6%, novelty has lost about 10%. In fact, it is easy for a graph-based method to generate 100% valid and high novelty molecules [1].

3. The authors state that LLMs may struggle to generate chemically valid or meaningful molecules without proper representation. In fact, in the work of using LLM for molecule generation, using SMILES, validity can reach over 99%, and novelty can reach over 90% depending on the dataset [2].

4. Further explanation is needed on how to restrict the generated tokens. There is a lack of description of the specific implementation details, such as how token screening is conducted, which tokens are considered valid, and which are not.

5. The ablation experiments demonstrate that the validity of the generated molecules heavily relies on token constraining. Without this, the validity of the generated molecules drops to only 41.60%. This result also confirms that compared to using SMILES or other molecular serialization representation methods, JSON does not offer any advantages, but rather increases the model's complexity. In contrast, the work referenced in [2] did not use a token constraining method, yet the validity of the generated molecules exceeded 99%.

[1] Geng, Zijie, et al. "De novo molecular generation via connection-aware motif mining." arXiv preprint arXiv:2302.01129 (2023).

[2] Zhou, Peng, et al. "Instruction Multi-Constraint Molecular Generation Using a Teacher-Student Large Language Model." arXiv preprint arXiv:2403.13244 (2024).

**Questions:**

See weaknesses.

---

### Note · Authors · 2024-11-22

I have read and agree with the venue's withdrawal policy on behalf of myself and my co-authors.